# Bibliometric Analysis of Research on the Use of the Nine Hole Peg Test

**DOI:** 10.3390/ijerph191610080

**Published:** 2022-08-15

**Authors:** Gema Moreno-Morente, Miriam Hurtado-Pomares, M. Carmen Terol Cantero

**Affiliations:** 1Department of Surgery and Pathology, Miguel Hernández University, 03550 Alicante, Spain; 2Grupo de Investigación en Terapia Ocupacional (InTeO), Miguel Hernández University, 03550 Alicante, Spain; 3Department of Behavioral Sciences and Health, Miguel Hernández University, 03550 Alicante, Spain

**Keywords:** nine hole peg tests, dexterity, upper extremity, hand function, evaluation

## Abstract

Manual dexterity is essential for performing daily life tasks, becoming a primary means of interaction with the physical, social, and cultural environment. In this respect, the Nine Hole Peg Test (NHPT) is considered a gold standard for assessing manual dexterity. Bibliometrics is a discipline that focuses on analyzing publications to describe, evaluate, and predict the status and development trends in certain fields of scientific research. We performed a bibliometric analysis to track research results and identify global trends regarding the use of the NHPT. The bibliographic data were retrieved from the Web of Science database and then analyzed using the Bibliometrix R package, resulting in the retrieval of a total of 615 publications from 1988 to 2021. Among the 263 journals investigated, the most prolific were the Multiple Sclerosis Journal, Clinical Rehabilitation, and Multiple Sclerosis and Related Disorders. North America and Europe were the areas with the highest production of publications, with the United States (*n* = 104) ranking first in terms of the number of publications, followed by the United Kingdom (*n* = 62) and Italy (*n* = 62). The analysis of keywords revealed that there were two main lines of research, with one related to the study of recovery and disability of the upper limbs caused by certain diseases and another related to the study of reliability and validity. Structured information can be useful to understand the research trajectory and the uses of this tool.

## 1. Introduction

Bibliometrics is a discipline [1] that focuses on the study of publications to describe, evaluate, and predict the status and development trends in certain fields of scientific research [2]. It provides indicators to measure scientific production and quality [3] and allows for an area of research to be studied through the growth and citation of publications, active authors, countries and institutions, international collaboration, and the frequency of terms, which in turn opens the door for other future lines of research. In this context, this study focused on The Nine Hole Peg Test (NHPT) [4]; the gold standard and the most commonly used tool for assessing manual dexterity in a wide range of clinical and research areas [5].

Dexterity has been defined as “the fine, voluntary movements used to manipulate small objects during a specific task, as measured by the time to complete the task” [6]. Manual dexterity may be determined by individual factors, such as age, gender, educational level, and hand dominance [7,8]. In addition, during one’s lifetime, hand dexterity may be altered by hand injuries or certain pathologies such as a stroke, multiple sclerosis, or Parkinson’s disease, which limit participation in daily activities, such as self-care tasks, typing on a computer keyboard, messaging on a cell phone, completing work related tasks, and engaging in leisure activities [9,10,11]. Therefore, assessing dexterity is critical because it is a central component of hand functioning [8] and is considered essential for a person to successfully perform tasks in daily life, work, school, play, and leisure activities [12].

The NHPT was originally introduced by Kellor et al. [13] in 1971 as a measure of dexterity. In their report, they provided approximate dimensions for the test and the general procedures for its administration. In 1985, Mathiowetz et al. [14] added detailed test instructions and adult normative values according to hand, sex, and age. The test consists of a plastic console with a shallow round dish for the pegs at one end and a nine-hole peg-board at the opposite end. It measures fine motor dexterity in terms of the number of seconds (completion time) a subject takes to place the nine pegs in the pegboard and then remove them [13]. The board should be placed in front of the person, and the test includes an initial practice to familiarize the individual with the procedures and assesses fine motor dexterity with the dominant and non-dominant arm [14].

The NHPT is widely used to assess manual dexterity in populations affected by hand dysfunction due to hand injuries, chronic and neurological diseases (stroke [15,16,17], multiple sclerosis [18,19,20], Parkinson’s disease [1], or Charcot–Marie–Tooth disease [13]. A large number of studies reported that the NHPT is a valid and reliable tool for assessing manual dexterity and support its potential usefulness as a sensitive measure of change, suggesting that it is a good instrument for improving diagnosis, detecting symptoms as well as for planning and monitoring rehabilitation interventions.

Despite the current relevance of manual dexterity as an important indicator of independence in occupations and the role of the NHPT as an outcome measure in this field, there is no evidence of a comprehensive review of this subject in the literature. Therefore, such a review could be very useful to scientific production in this field from an integrative perspective and would thereby provide visibility of the wide use of the NHPT. In this context, the aim of this study was to review the available evidence that reports the use of the NHPT for evaluating manual dexterity in order to learn about its research trajectory while considering the following specific scientific production indicators: years of evolution, countries and institutions, journals and categories distribution, representative authors and article citations, and the frequency and high frequency of key words.

## 2. Materials and Methods

### 2.1. Search Strategy and Data Extraction

The Web of Science (WoS) database was selected to perform the literature search for all published articles on the NHPT, covering the period from its inception to 31 December 2021 and with no language limitation. The search was conducted using the terms “nine hole peg test” and “nhpt” and “nine hole” and “9hpt” for the topic field, which included title, abstract, author keywords, and keyword plus terms. All references indexed and published until December 2021 were included in the analysis. In order to identify possible publications that were not related to the field and to minimize any errors in the data provided by the database, all retrieved documents were examined. The following data were extracted from each publication: title, journal, article type, author names and affiliations, keywords, date of publication, research area, and abstract. The search strategy was as follows: ((TITLE-ABS-KEY (nine hole peg test) OR TITLE-ABS-KEY (nhpt) OR TITLE-ABS-KEY (nine hole) OR TITLE-ABS-KEY (9hpt)) 9hpt) (Figure 1).

### 2.2. Data Analysis and Visualization

The data from the bibliographic search were exported into BIB format from the WOS database. The bibliometric analysis was conducted using R software version 3.6.2 (R Foundation for Statistical Computing, Vienna, Austria; http://www.r-project.org (accessed on 14 March 2022)) in the Bibliometrix R package. Based on annual scientific production, this package provides a set of tools for quantitative research in bibliometrics and scientometrics. Scientific production and collaboration were calculated and ranked based on the most cited papers, authors, countries/regions and institutions, journals, and the most used terms. The information on countries and institutions was obtained from the first author’s country affiliation, and MapChart (https://mapchart.net (accessed on 1 April 2022)) was used to create a world map to display the geographical distribution of publications on the NHPT. The type of documents and general categories were obtained using the intrinsic function of the WOS. The influence and quality of journals were also measured using the impact factor obtained from the latest Journal Citation Reports (JCR) (2020) created by Clarivate Analytics. The VOSviewer program (http://www.vosviewer.com/ (accessed on 1 April 2022)) [21] was used for data visualization, which creates scientific landscapes and networks based on keywords and keywords Plus.

## 3. Results

### 3.1. Evolution over the Years

The analysis returned a total of 615 articles. The first article was published in 1988 and the number of articles in this research area increased by 13.1% per year. Although annual publications were initially lower, in 2008 research production began to experience a progressive data growth which continuously improved since then. Annual research production on NHPT is shown in Figure 2.

### 3.2. Countries and Institutions

The 615 publications originated from 47 countries across five continents (Figure 3). Of these, 12 countries had only 1 publication, 19 countries had 2–9 publications, and 16 had at least 10 publications on NHPT.

Table 1 shows the 20 most productive countries, with the United States ranking first with Total Number of Documents (TND) of 104 (16.9%), followed by the United Kingdom and Italy (*n* = 62, 10.1%), Turkey (*n* = 47, 7.64%), the Netherlands (*n* = 42, 6.83%) and Germany (*n* = 41, 6.67%). Regarding single country publications (SCP), the United States presented the highest number of publications by authors from the same country (*n* = 80), followed by Italy (*n* = 49), Turkey (*n* = 47), the United Kingdom (*n* = 45), the Netherlands (*n* = 29) and Germany (*n* = 28). On the other hand, those with the highest number of publications with authors from different countries (MCP = Multiple country publications) were the United States (*n* = 24), the United Kingdom (*n* = 17) and Italy, Germany and the Netherlands (*n* = 13). In relative terms, the highest values of cross-country productivity or the MCPRatio (MCPª) index were found for Belgium (MCP = 0.5, TND = 14) and Switzerland (MCP = 0.5, TND = 22) followed by Canada (MCP = 0.39, TDN = 26), Germany (MCP) = 0.32, TND = 41), Australia (MCP = 0.31, TND = 16) and the Netherlands (MCP = 0.31, TND = 42).

Finally, Figure 4 shows the 20 most collaborative countries in terms of number of bilateral relationships created. A stronger collaboration relationship between two countries is shown with a thicker line and a larger circle size indicates a higher number of international collaborative projects. From this perspective, the USA (*n* = 112) was the centre of collaboration in this field and its global international activity was greater than that of other countries. This was followed by the UK (*n* = 83), Germany (*n* = 60), Italy (*n* = 56), the Netherlands (*n* = 49), Switzerland (*n* = 22), and Canada (*n* = 21).

Table 2 shows the 20 most prolific institutions to have published articles about the NHPT. The most active was the University of Wisconsin with 50 articles, (8.13%) followed by Dokuz Eylul University with 23 (3.74%), and Urije University Amsterdam and Washington University with 22 articles each (3.58%). Of the 615 articles retrieved, the 20 most prolific institutions published 368 (59.8%) articles, 13 (70%) of which were located in Europe.

### 3.3. Journals and Category Distribution

In relation to journals, all of the 615 retrieved articles were based on 263 sources (article types, journals, books or proceeding papers among other types of scientific documents). Regarding the type of document, most of them were research articles (*n* = 558, 90.73%) followed by proceeding papers (*n* = 30, 4.88%) or reviews (*n* = 11, 1.79%) and meeting summaries, editorial issues, early access, letters to editors, book reviews and notes (*n* = 16, 2.6%). As research articles are the most frequent type of document, journals are the most prominent sources of publication.

The 20 most prolific journals focus on Rehabilitation and Neurology. The ones that published the most articles about NHPT were the Multiple Sclerosis Journal (37 publications, 6.02%), followed by Clinical Rehabilitation (*n* = 20, 3.25%), and Multiple Sclerosis and Related Disorders (*n* = 20, 3.25%) (Table 3).

The general category distribution of the articles included in this study between 1988 and 2021 are shown in Figure 5. Most of the articles belong mainly to the clinical neurology and neuroscience category (*n* = 264, 42.93%; *n* = 205, 33.33%, respectively). A significant number of articles were also published in categories or subject areas such as rehabilitation (*n* = 166, 27%); surgery (*n* = 42, 6.83%); sports science (*n* = 32, 5.2%) and orthopedics (*n* = 27, 4.39%).

### 3.4. Representative Authors and Citations

There were 6 authors of single-author articles and 2851 authors of multi-author articles that contributed to the 615 publications. A total of 5 authors published more than 10 articles, 30 authors between 5 and 9 articles and 421 authors published between 2 and 4 articles and the number of authors of a single article, also known as the transience index, was 2401, representing 84.04% of the total number of authors.

The details of the most prolific authors are presented in Figure 6. Cattaneo D., Feys P. and Uitdehaag BMJ (*n* = 12 articles) were the most productive authors in this field, followed by Polman C.H. and Solaro C. with 10 articles, Bertoni R. and Liepert J. with 9 articles, followed by Lamers I., Kragt J. and Vanbellingen T. with 8 articles each.

The co-authorship analysis indicated an average of 5.93 co-authors per article and a collaboration rate of 4.68; that is to say, the total number of authors of articles with several authors (*n* = 2851) divided by the total number of articles with several authors (*n* = 609).

Citation per author was measured by the H-Index and the data were analyzed to find many times an author was cited in NHPT publications. There were 290 authors without citations (10.15%), 188 authors were cited at least once (6.58%), 834 authors had between 2 and 9 citations (29.2%), 948 authors had between 10 and 49 citations (33.2%) and 597 had more than 50 citations (20.9%). Pollman C.H., with 10 publications, was the most influential author with 622 citations, followed by Miller D.H. (604 citations) and Liepert J. (535 citations) (Table 4).

If we look at the h-index of the 20 most cited authors, Miller D.H., who was cited a total of 604 times, had an h-index of 6 compared to Kragt J.J., who with 285 citations had an h-index of 8. The duration of academic career, measured by the M-index, showed us that Vanbelligen (0.78), Bertoni R. (0.63), Lamers I. (0.58), Tachinno A. and Uitdehaag, B.M.J. (0.50) were the authors with the highest growth in terms of their scientific production (Table 4).

With respect to article citations about the NHPT, the 615 articles available generated a total of 15,368 citations. A total of 47 (7.64%) articles had at least one citation, while 82 (13.3%) articles had no citations. The top 20 cited papers are listed in Table 5. All of them were cited more than 100 times and the highest citation number was for the article titled “Its Occurrence and Association with Motor Impairments and Activity Limitations” (Sommerfeld et al., 2004), with 437 citations and an average of 23.00 citations per year.

### 3.5. Frequency of Key Words

A total of 1361 author keywords were retrieved. The most used author keywords occurred from a minimum of eleven to a maximum of 155 times. Keywords represent the highly concentrated content of literature research, indicating the focus of the research field. We analyzed the most frequently used author keywords and the keywords associated with the manuscript by the WOS database following the bibliometric studies index (Figure 7 and Figure 8, respectively). In these figures, the largest diameter of the nodes represents the highest frequency of the keyword, while the largest thickness of the path lines represents the proximity of the co-occurrence relationships. In Figure 7, the author keyword link structure revealed five different clusters represented by different colors. The author’s keywords with the highest frequency were multiple sclerosis (*n* = 155), stroke (*n* = 79), rehabilitation (*n* = 77), upper extremity (*n* = 40), dexterity (*n* = 34) and hand function (*n* = 26).

In addition, a total of 1464 plus keywords were found. The most frequently used plus keywords found were a minimum of 27 to a maximum of 111 times. As shown in Figure 8, five groups resulting from the network analysis of keyword co-occurrences were identified as the knowledge structure of NHPT research. The most used keyword plus was reliability (*n* = 111), followed by disability (*n* = 88), impairment (*n* = 69), recovery (*n* = 68) and validity (*n* = 64).

## 4. Discussion and Conclusions 

This study presented a bibliometric analysis that included existing research on the use of the NHPT manual dexterity test. The analysis showed a total of 615 publications with an increase in the number of publications in recent years. Most documents were published between 2015 and 2020, with a significant increase in 2008, following a general trend of constant growth to date. We postulate that this considerable increase in the use of NHPT in research may be due to advances in the study of certain neurodegenerative and/or neurological diseases for their early diagnosis and treatment [40]. Another reason could be the growing number of people with these diseases [41], mainly due to the increase in life expectancy of the general population [42,43]. In fact, most of the articles published on NHPT usually encompass topics related to neurological issues in general, and multiple sclerosis in particular. Research was carried out with articles (*n* = 558, 90.73%) published in journals that focus on clinical neurology, neuroscience and rehabilitation or more specifically, in journals that mainly cover research findings related to multiple sclerosis or other diseases of neurological origin. The extensive research on these topics could also be explained by the fact that the NHPT is considered a “gold standard” [19] for evaluating manual dexterity, a function that is affected by many diseases of the central nervous system.

The three most prolific journals found in our results (Multiple Sclerosis Journal, Clinical Rehabilitation, and Multiple Sclerosis and Related Disorders) were written in English. According to the 2020 Journal Citation Report (JCR), the first two journals are included in the first quartile for Neurosciences, Clinical neurology and Rehabilitation and the last journal is included in the second quartile for Clinical Neurology. The Multiple Sclerosis Journal and Multiple Sclerosis and Related Disorders focus on aspects related to multiple sclerosis and diseases associated with the central nervous system. Clinical Rehabilitation covers the entire field of disability and rehabilitation, giving priority to research articles that describe the effectiveness of therapeutic interventions and the evaluation of new techniques in this field. The Multiple Sclerosis Journal has one of the 20 most cited articles on NHPT (Feys P., 2017) where the NHPT is endorsed as the optimal metric for measuring the impact of Multiple Sclerosis on upper extremity function. This could be due to the fact that Multiple Sclerosis represents an important study area where the assessment of dexterity with the NHPT is providing satisfactory clinical results [5,43,44]. In addition, it should be noted that in 1997, the National MS Society Clinical Outcomes Assessment recommended the use of the NHPT as an upper extremity outcome measure in multiple sclerosis [45,46]. Two years later, the Multiple Sclerosis Function Composite (MSFC) was published, which includes the NHPT as an outcome measure. Since 1999, the NHPT has frequently been included in the clinical practice of MS and in investigations on this disease [5]. This, in turn, supports the results found in this study of how the use of the NHPT has increased considerably in recent decades. 

This trend is also confirmed by the analysis of keywords such as multiple sclerosis, stroke, rehabilitation, upper extremity, hand dexterity and function, showing that neurological pathologies and rehabilitation of the upper extremity have been the main focuses of research. At the same time, researchers have also paid attention to issues related to the reliability and validity of measurement tools. In this respect, the NHPT has demonstrated appropriate measurement properties in healthy children and adults with neurological conditions [47].

From a distribution perspective, the USA has been the centre of collaboration and its cooperative strength has been much greater than that of other countries. The United States has also led other countries in terms of the total number of publications (104 publications, 16.9%). This is not surprising since the United States leads the world in scientific production [48]. When it comes to institutions, the University of Wisconsin ranked first in the total number of publications. In fact, Mathiowetz [14], who authored the NHPT measure and published the normative data, is from the University of Wisconsin. Despite this, 14 (70%) of the 20 most productive institutions were located in Europe. If we analyze the documents included in this study by decades in terms of their distribution between countries, we observe that in the first decade the vast majority of the studies were carried out in the United States, the country where the instrument was designed and validated. However, in the following two decades, there was an increase in papers published by European countries. More specifically, in the second decade many studies were carried out in Germany and Holland, and in the most recent decade, Italy also appeared as one of the most productive countries. The latter data may be related to the validation of the NHPT as a measure of dexterity in myotonic dystrophy type 1 [49].

The vast majority of the 20 most prolific authors started publishing after 2002 and the m-index showed that some of them have had a high level of scientific production in a relatively short period of time, indicating the growing progress of research on the use of NHPT. These data coincide with the publication of the new NHPT guidelines for school-aged children [50] and adults [4] using the current marketed version, which also endorses the NHPT as an effective screening tool for fine motor skills of children and adults in addition to supporting the original standards [14].

This study presents several potential limitations. Firstly, the search was only carried out on the WOS, since it is a database with a wide range of scientific journals [51]. However, we are also aware that those articles that do not appear in this database were not included in our results. Although the search results were the same when the words “9-HPT” and “Nine-Hole Peg Test” were also included, we decided not to include them in our search, which could have resulted in a possible inclusion bias for comparison with other bibliometric studies in the same area. The procedure for data extraction and transformation using WoS and Bibliometrix might lead to erroneous results or missing data; therefore, the bibliographic information was reviewed manually by G.M.M.

### Future Research and Practical or Clinical Applications

It should be noted that this is the first bibliometric analysis on the use of the NHPT; and therefore, it provides data on the research categories where it has been studied, the main authors and publication journals, and the research trajectory in this field. The NHPT is an assessment tool used to measure manual dexterity and was designed and validated in a healthy adult population. However, it is important to note that over the years its use in the evaluation of manual dexterity was extended to people with various neurological diagnoses. Currently, research on the applicability of the NHPT comprises a wide range of different areas of study, although its main focus is on neurology and rehabilitation. The studies found and included in this analysis indicate the trend of its use in neurodegenerative diseases such as multiple sclerosis or neurological diseases with stroke. It is also used, but to a lesser extent, to screen other diseases related to the involvement of the peripheral nerves of the upper limbs such as Charcot–Marie–Tooth disease or carpal tunnel syndrome.

Its easy application, the simplicity of its instructions, as well as its adequate psychometric properties for the evaluation of manual dexterity have made it a “gold standard” to measure said ability. Its applicability has also been extended to research studies related to the study of the validity and reliability of other similar evaluation instruments.

The beginnings of research on the NHPT were carried out mainly in the US. However, over the years, it has spread much more to European countries such as Germany, Italy or Spain, even though it is a tool that has not been validated in the vast majority of these countries. This study may be of great use to scientific researchers, and also to clinicians in this field interested in developing an effective measurement instrument for the population affected by hand dysfunction caused by neurological disease. In addition, it could also contribute to its adaptation and validation in those countries where it has not yet been validated.

## Figures and Tables

**Figure 1 ijerph-19-10080-f001:**
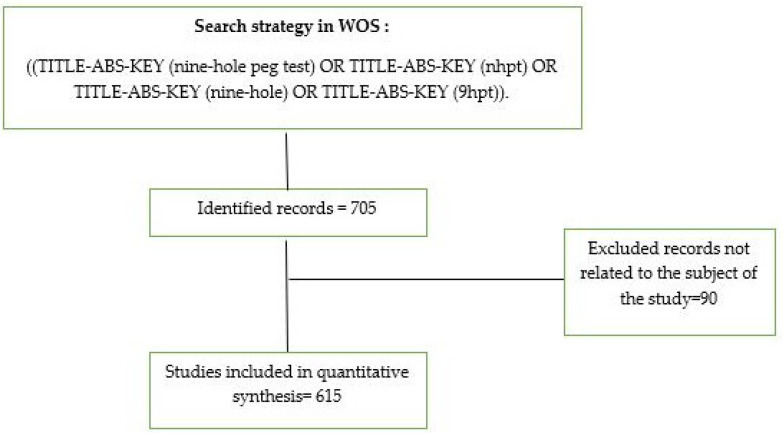
Search process and selection of publications on the NHPT.

**Figure 2 ijerph-19-10080-f002:**
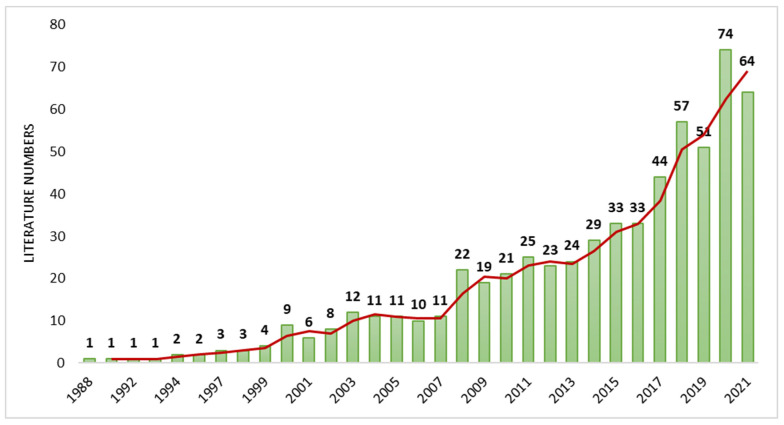
Annual distribution of publications on NHPT.

**Figure 3 ijerph-19-10080-f003:**
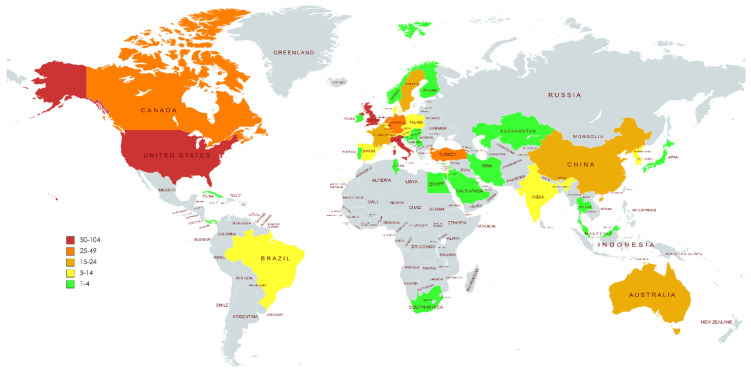
Geographical distribution map of the publications on NHPT.

**Figure 4 ijerph-19-10080-f004:**
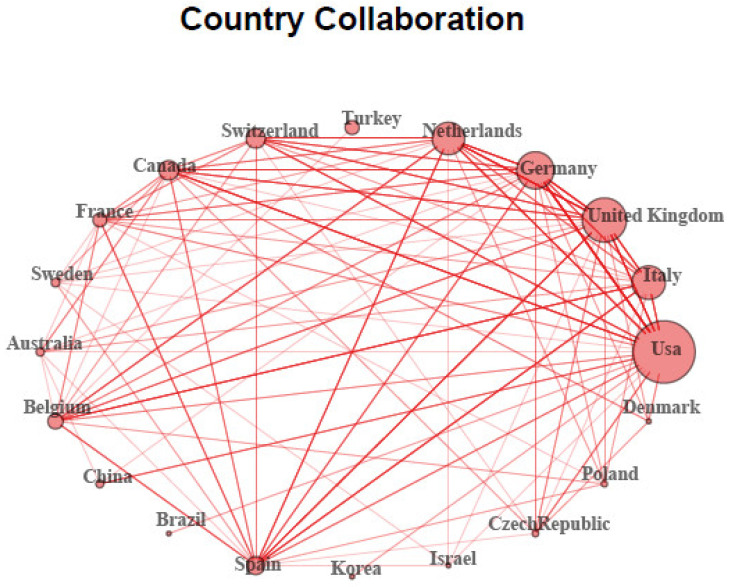
Country collaboration network map (thicker lines: stronger collaborations; larger circle size: higher number of international collaborative projects).

**Figure 5 ijerph-19-10080-f005:**
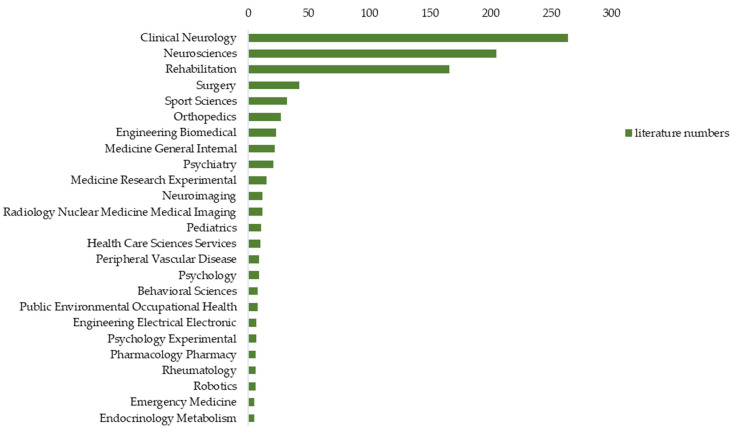
Publications on NHPT from 1988 to 2021 by general categories.

**Figure 6 ijerph-19-10080-f006:**
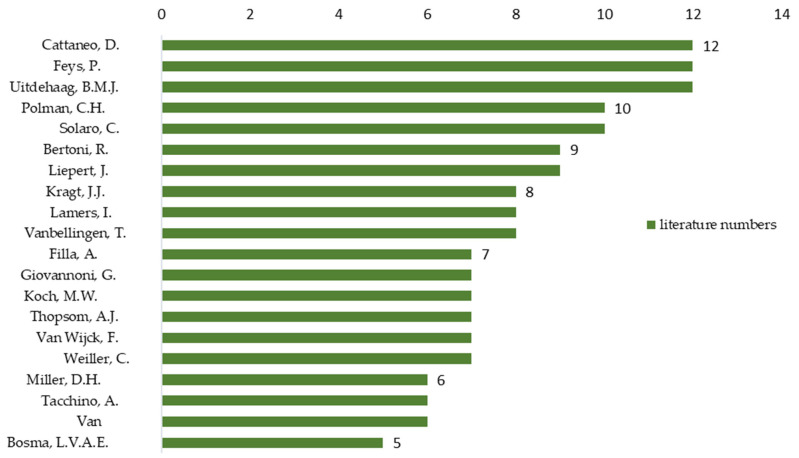
Top 20 most prolific authors in publishing papers on NHPT.

**Figure 7 ijerph-19-10080-f007:**
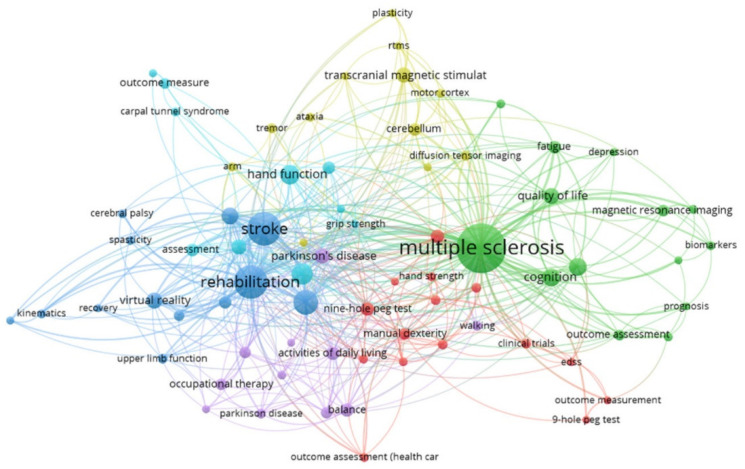
Author keyword co-occurrences network map.

**Figure 8 ijerph-19-10080-f008:**
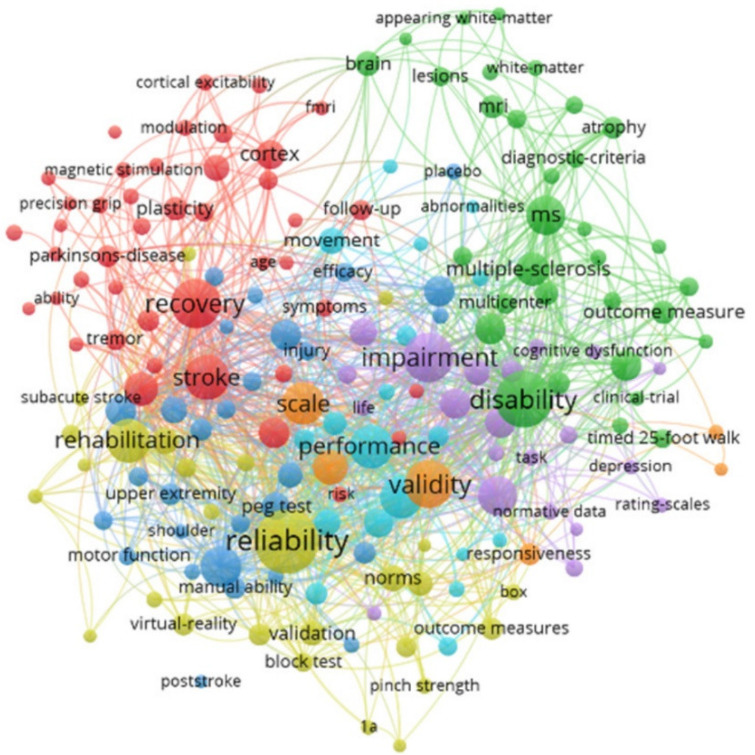
Keyword plus terms co-occurrences network map.

**Table 1 ijerph-19-10080-t001:** Top 20 prolific publishers on NHPT by country.

Countries	TND	% ^a^	SCP	MCP	MCP Ratio ^b^
USA	104	16.7	80	24	0.231
Italy	62	10.1	49	13	0.210
United Kingdom	62	10.1	45	17	0.274
Turkey	47	7.64	47	0	0.000
Netherlands	42	6.3	29	13	0.310
Germany	41	6.7	28	13	0.317
Canada	26	4.2	16	10	0.385
Switzerland	22	3.6	11	11	0.500
China	18	2.93	13	5	0.278
Sweden	18	2.93	16	2	0.111
France	17	2.8	13	4	0.235
Australia	16	2.6	11	5	0.312
Belgium	14	2.3	12	2	0.500
Brazil	14	2.3	12	2	0.143
Israel	11	1.8	9	2	0.182
Korea	10	1.63	10	0	0.000
Spain	9	1.5	9	0	0.000
Czech Republic	8	1.3	7	1	0.125
Denmark	8	1.3	7	1	0.125
Poland	8	1.3	8	0	0.000

TND: Total number of documents; SCP: single country publications; MCP: multiple country publications; ^a^: Percentage calculated out of the 615 retrieved documents; ^b^: Multiple country publication ratio was calculated as MCP divided by the total of published documents per country.

**Table 2 ijerph-19-10080-t002:** Top 20 institutions for publications on NHPT sorted by total number of articles.

Research Institute	Country	Number of Articles	% ^a^
University of Wisconsin	USA	50	8.13
Dokuz Eylul University	Turkey	23	3.74
Vrije University Amsterdam	Holland	22	3.58
Washington University	USA	22	3.58
University Basel	Switzerland	21	3.41
Charite University Berlin	Germany	20	3.25
University Genoa	Italy	19	3.09
Karolinska Institute	Stockholm	17	2.76
Institute of Neurology	London	16	2.6
Mcgill University	Canada	16	2.6
UCL Queen Square Institute of Neurology	London	16	2.6
Radboud University Nijmegen	Netherlands	15	2.44
University of Calgary	Canada	15	2.44
Vrije University Amsterdam Med Ctr	Holland	15	2.44
Gazi Üniversitesi	Turkey	14	2.28
University of Bern	Switzerland	14	2.28
University of Groningen	Netherlands	14	2.28
Tel Aviv University	Israel	13	2.11
University Hospital Bern	Switzerland	13	2.11
University of Toronto	Canada	13	2.11

Abbreviations: MED: Medicine; MED CTR: Faculty of Medicine; ^a^: Percentage calculated out of the 615 retrieved articles.

**Table 3 ijerph-19-10080-t003:** Top 20 most prolific publishers on NHPT by journal.

Rank	Journals	Number of Articles (%) ^a^
1st	Multiple Sclerosis Journal	37(6.02)
2nd	Clinical Rehabilitation	20(3.25)
	Multiple Sclerosis and Related Disorders	
4th	Journal of Neurology Neurosurgery and Psychiatry	16(2.6)
5th	Neurology	15(2.44)
	Archives of Physical Medicine and Rehabilitation	
7th	European Journal of Neurology	13(2.11)
	Journal of Neuroengineering and Rehabilitation	
9th	Journal of Neurology	12(1.95)
10th	Neurorehabilitation and Neural Repair	11(1.79)
11th	Journal of Neurological Sciences	10(1.63)
12th	Neurorehabilitation	9(1.46)
13th	Clinical Neurophysiology	8(1.3)
	Cerebelum	
14th	Disability and Rehabilitation	7(1.14)
	Multiple Sclerosis	
17th	Annals of Rehabilitation Medicine-Arm	6(0.98)
	Brain	
	Frontiers in Neurology	
	Journal of Hand Therapy	

^a^: Percentage calculated out of the retrieved 615 articles.

**Table 4 ijerph-19-10080-t004:** Top 20 most cited authors publishing on NHPT by number of citations.

Author	H-Index	G-Index	M-Index	TC	NP	YFP
Uitdehaag, B.M.J.	9	12	0.50	479	12	2005
Cattaneo, D.	6	11	0.43	261	11	2009
Feys, P.	9	10	0.43	527	10	2002
Polman, C.H.	10	10	0.48	622	10	2002
Liepert, J.	7	9	0.32	535	9	2001
Bertoni, R.	5	8	0.63	180	8	2015
Kragt, J.J.	8	8	0.47	285	8	2006
Solaro, C.	4	8	0.31	79	8	2010
Vanbellingen, T.	7	8	0.78	150	8	2014
Giovannoni, G.	5	7	0.24	336	7	2002
Lamers, I.	7	7	0.58	479	7	2011
Thompson, A.J.	7	7	0.33	439	7	2002
Weiller, C.	7	7	0.32	458	7	2001
Koch, M.W.	4	6	0.44	52	6	2014
Miller, D.H.	6	6	0.29	604	6	2002
Van Wijck, F.	4	6	0.27	220	6	2008
Bosm, L.V.A.E.	5	5	0.36	123	5	2009
Tacchino, A.	4	5	0.50	43	5	2015
Van	5	5	0.21	243	5	1999
Filla, A.	4	4	0.27	211	4	2008

Abbreviations: TC, total citations; NP, number of publications; YFP, year of first indexed publication.

**Table 5 ijerph-19-10080-t005:** Top 20 articles cited on NHPT from inception to 2021.

Ranking	Author	Title	Year	Journal	TC	TCy	IF
1st	Sommerfeld, D.K., et al. [22]	Its Occurrence and Association with Motor Impairments and Activity Limitations	2004	Stroke	437	23.00	7914
2nd	Grice, K.O., et al. [4]	Adult norms for a commercially available Nine Hole Peg Test for finger dexterity	2003	AJOT	345	17.25	2246
3rd	Cohen, J.A., et al. [23]	Benefit of interferon β-1a on MSFC progression in secondary progressive MS	2002	Neurology	293	13.95	9910
4th	Chen, H.M., et al. [24]	Test–retest Reproducibility and Smallest Real Difference of 5 Hand Function Tests in Patients with Stroke	2009	NNR	236	16.86	3919
5th	Lublin, F., et al. [25]	Oral fingolimod in primary progressive multiple sclerosis (INFORMS): a phase 3, randomised, double-blind, placebo-controlled trial	2016	Lancet	230	32.86	79323
6th	Goodkin, D.E., et al. [26]	Upper extremity function in multiple sclerosis: improving assessment sensitivity with box-and-block and nine-hole peg tests	1988	APMR	217	6.20	966
7th	Heald, A., et al. [27]	Longitudinal study of central motor conduction time following stroke: 2. Central motor conduction measured within 72 h after stroke as a predictor of functional outcome at 12 months	1993	Brain	188	6.27	13501
8th	Duncan, R.P., et al. [28]	Randomized Controlled Trial of Community-Based Dancing to Modify Disease Progression in Parkinson Disease	2012	NNR	186	16.91	3919
9th	Leary, S.M., et al. [29]	Interferon β-1a in primary progressive MS An exploratory, randomized, controlled trial	2003	Neurology	179	8.95	9910
10th	Henry, R.G., et al. [30]	Regional grey matter atrophy in clinically isolated syndromes at presentation	2008	JNNP	176	11.73	10283
11th	Pareyson, D., et al. [31]	Ascorbic acid in Charcot–Marie–Tooth disease type 1A (CMT-TRIAAL and CMT-TRAUK): a double-blind randomised trial	2011	Lancet Neurol	165	13.75	44182
12th	Vaney, C., et al. [32]	Efficacy, safety and tolerability of an orally administered cannabis extract in the treatment of spasticity in patients with multiple sclerosis: a randomized, double-blind, placebo-controlled, crossover study	2004	Mult Scler-a	162	8.53	6312
13th	Petzold, A., et al. [33]	Markers for different glial cell responses in multiple sclerosis: clinical and pathological correlations	2002	Brain	155	7.38	13501
14th	Schimrigk, S., et al. [34]	Oral fumaric acid esters for the treatment of active multiple sclerosis: an open-label, baseline-controlled pilot study	2006	Eur. J. Neurol.	153	9.00	6089
15th	Feys, P., et al. [5]	The Nine-Hole Peg Test as a manual dexterity performance measure for multiple sclerosis	2017	Mult. Scler. J.	152	25.33	6312
16th	Fisk, J.D., et al. [35]	A comparison of health utility measures for the evaluation of multiple sclerosis treatments	2005	JNNP	147	8.17	10283
17th	Liepert, J., et al. [36]	Motor cortex plasticity during forced-use therapy in stroke patients: a preliminary study	2001	J. Neurol.	138	6.27	4849
18th	Merkies, I.S.J., et al. [37]	Psychometric evaluation of a new sensory scale in immune-mediated polyneuropathies	2000	Neurology	137	5.96	9910
19th	Santisteban, L., et al. [38]	Upper Limb Outcome Measures Used in Stroke Rehabilitation Studies: A Systematic Literature Review	2016	Plos One	135	19.29	3240
20th	Holmqvist, L.W., et al. [39]	A Randomized Controlled Trial of Rehabilitation at Home After Stroke in Southwest Stockholm	1998	Stroke	135	5.40	7914

Abbreviations: TC, total citations; TCy, total citations per year; IF, impact factor (Journal Citations Report 2020); AJOT: American Journal of Occupational Therapy; J. Clin. Investig.: The Journal of Clinical Investigation; NNR: Neurorehabilitation & Neural Repair; APMR: The Archives of Physical Medicine and Rehabilitation; JNNP: Journal of Neurology, Neurosurgery and Psychiatry; Mult. Scler-a: Multiple Sclerosis Journal; Eur. J. Neurol.: The European Journal of Neurology; J. Neurol.: The Journal of Neurology; Mult. Scler. J.: Multiple Sclerosis Journal.

## Data Availability

Not applicable.

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
