# Peer review of "Bibliometric Analysis of Research on the Use of the Nine Hole Peg Test"

_ijerph, 2022, doi:10.3390/ijerph191610080_

Round 1
Reviewer 1 Report
Thank you very much for allowing me to read this manuscript. Here are my comments and recommendations:
· Remember that scientific reports are often written in an impersonal mode. English should be reviewed.
· Although the information is sufficient, I think the introduction could be improved if aspects of manual dexterity and the test were covered in more detail.
· Are the authors sure that bibliometrics is a discipline?
· Please use dots to delimit decimals.
· The discussion section is the results commented.
· It is clear to me that this is the first paper of its kind using this tool, but I believe that its usefulness and practical applications should be made more evident in the manuscript.
Author Response
Dear Mr / Mrs
We welcome your suggestions to improve our work. We have tried to point out all the changes introduced as you will see below. We also attach the document with all the changes introduced according to your recommendations and/or suggestions.
INTRODUCTION
I think the introduction could be improved if aspects of manual dexterity and the test were covered in more detail.
We added more specific information about the studies referenced between lines 39 and 48 and between lines 53 and 59 to improve the introduction section.
Are the authors sure that bibliometrics is a discipline?
We assume that bibliometrics is a discipline according the works by Vogel and Guttel (2013) and Quevedo and Lopez-Lopez (2010) where they state that: "In this sense, bibliometrics, a specialized technique in the study of productivity science, focuses on the quantitative analysis of the scientific literature published on a subject” (Vogel and Guttel, 2013).”This discipline applies the principles of scientific positivism to publications, offering in numerical terms the analysis of productivity indices” (Quevedo and López -López, 2010).These references are included in our work. (line 30 and 32).
Please use dots to delimit decimals.
We have changed the decimals throughout the text.
DISCUSSION
The discussion section is the results commented. It is clear to me that this is the first paper of its kind using this tool, but I believe that its usefulness and practical applications should be made more evident in the manuscript.
Bibliometric analysis allows to demonstrate a comprehensive overview of the field study of NHPT. Our study focused on: 1) to identify publications and citation trends, 2) to explore the distribution of paper types, 3) to recognize the most relevant countries/regions, affiliations and authors, and 4) to reveal relevant thematic features by analysing publication abstracts and titles with the use of word Keys and topics. The results are relevant because they highlight several research hotspots and emerging topics. Then, discussion section shows overall the quantitative data and bilbliometric review trying to approach to study area of NHPT. Other kind of study such as a systematic review will allow us to analyse more qualitative specific content of NHPT. However, we have included in the article (Discussion section) a new section about future researches and clinical or practical applications that could emerge from the results of this quantitative study. This information can be useful for researches and clinical practice in health settings.

Reviewer 2 Report
The authors report on bibliometric analysis performed for the the Nine Hole Peg Test 13 (NHPT), which is considered a gold standard for assessing manual dexterity. The analysis included 615 publications between the years 1988-2021. The conclusions are that the number of publications has increased throughout the years, and that they study is focused on neurology and rehabilitation. Other findings regarding publishing countries and institutions are reported.
The paper is interesting and nicely written. The study was performed correctly and reported adequately. However, I have trouble to find the scientific merit of the study. How knowing how many publications related to the topic were published throughout the years, from which institutions and countries, and from which areas, will help researchers in the future? I was expecting a study about the findings came out from the publications, advises regarding how to use the test correctly, and similar things. None of these was addressed in the study. The current results, unfortunately, do not add to our theoretical knowledge of the field, nor to the possible implications.
Some other comments:
1. Abstract: I miss a short explanation about the term "bibliometric", as well as a description about the results and their implications.
2. Introduction: A figure to demonstrate the test is missing. In addition, the rational behind performing the study should be explained - how is it going to assist and promote our future understanding?
3. Method: Did the authors performed any filtering based on the quality of the source? for example, eliminating predatory journals?
4. Results: The increasing in publications throughout the years should be analyzed based on the general increasing in publications, not only for this topic. The same should be performed to the rate of publishing countries and collaborations between the countries.
The collaborations should be explained, not only presented in a figure.
Author Response
Dear Mr / Mrs
We welcome your suggestions to improve our work. We have tried to point out all the changes introduced as you will see below. We also attach the document with all the changes introduced according to your recommendations and/or suggestions.
However, I have trouble to find the scientific merit of the study. How knowing how many publications related to the topic were published throughout the years, from which institutions and countries, and from which areas, will help researchers in the future? . I was expecting a study about the findings came out from the publications, advises regarding how to use the test correctly, and similar things. None of these was addressed in the study. The current results, unfortunately, do not add to our theoretical knowledge of the field, nor to the possible implications.
Bibliometric analysis allows to demonstrate a comprehensive overview of the field study of NHPT. This kind of studies focus on: 1) to identify publications and citation trends, 2) to explore the distribution of paper types, 3) to recognize the most relevant countries/regions, affiliations and authors, and 4) to reveal relevant thematic features by analysing publication abstracts and titles with the use of word Keys and topics. The results are relevant because they highlight several research hotspots and emerging topics. In addition, bibliometric studies offer a convenient information for studying collaboration in research, topics, top countries and so on. The results of bibliometric studies provide useful insights into the general status and trends of published scientific literature in relevant areas and the development trends of the areas (Chen, Lun, Yan, Hao, & Weng, 2019). Bibliometric analysis could be complemented by a systematic review for a more extend and meticulous approach including quantitative and qualitative information. However, bibliometrics is the most extensively practiced approach to trace the knowledge anatomy of a research field (Li, Wu, & Wu, 2017) and research topics (Blanco-Mesa, Merigó, & Gil-Lafuente, 2017) while systematic literature reviews are used to synthesize the contents of the literature, limit bias (Tranfield, Denyer, & Smart, 2003). In this work, we have focused on quantitative data and bilbliometric review trying to approach to study area of NHPT. We consider that including a systematic review in the same paper could be excessive, being in itself the object of study for another article. Despite all this, we have included in the article (lines between 393 and 425) a new section with information about clinical and practical applications that could emerge from the results of this study.
ABSTRACT
I miss a short explanation about the term "bibliometric", as well as a description about the results and their implications.
In the abstract we have included a short explanation about of bibliometrics between lines 14 and 16.
INTRODUCTION
A figure to demonstrate the test is missing. In addition, the rational behind performing the study should be explained - how is it going to assist and promote our future understanding?
We consider that including a figure to show the test is not necessary. It can be found on Internet, and it is not possible to publish without a permission from the company that markets it. In line 77 we explain that it is intended to know the widespread use of said measurement tool.
METHOD
Did the authors performed any filtering based on the quality of the source? for example, eliminating predatory journals?
We included all journals since we considered it is important to analyse and show quantitative indicators of. Perhaps if our objective had been to make a more detailed analysis of the applicability and adequacy of administration of the instrument, we would have made a selection of the most prestigious journals. But that exceeded the objective of the study.
RESULTS
The increasing in publications throughout the years should be analyzed based on the general increasing in publications, not only for this topic. The same should be performed to the rate of publishing countries and collaborations between the countries.
We agree that a substantial proportion of research collaborations and publications with other scientists have increased simultaneously through different disciplines in the last decades, producing an upward trend in the numbers of papers and ties between individuals focused on different topics studied (Steen, Casadevall, A., & Fang, 2016). This suggest that authors network in a research area revealed a variety of different types of relationships between members that results in a spread of scientific workers, researches and development activities around the world (Crane, 1972). In addition, as other authors have found (e.g., Katz and Hicks 1997; Rigby 2009; Guerrero Bote, Olmeda- Gómez, and Moya- Anegón 2013; Lancho Barrantes et al. 2012; Adams 2013), the international networks tend to produce more highly papers and collaborations between persons. But all of this circumstances are especially relevant when Occupational therapy and Physiotherapy expanded as a discipline for supporting the researches and therapies of neurodegenerative and/or neurological disease, their early diagnosis and treatment. This has also been our point of view in our bibliometric article that focus on the increasing o NHPT papers as a useful tool in neurology area and so we have tried to highlight.
The collaborations should be explained, not only presented in a figure.
We included the information about the collaborations between line 166 and 169.

Reviewer 3 Report
This paper provides a bibliometric analysis of research focusing on the Nine Hole Peg test, which is the gold standard for assessing manual dexterity. Specific comments regarding the manuscript are given below.
1. A critical point of the analysis is the identification of relevant papers. However, some key terms are not used when undertaking the literature search, e.g. “Nine-Hole Peg Test”, as mentioned within the discussion. However, the exclusion of this term could significantly affect the analysis, in particular as this term is shown as a prominent keyword within the literature, as illustrated within Figure 6. I would strongly recommend extending the literature search to ensure that all relevant papers are identified and included within the analysis.
2. When discussing Figure 1, the authors mention a growth in the number of publications from 2008. Some discussion on why this is would aid the reader, e.g. what paper(s) were published that year which created a shift in the research direction?
3. As the papers being analysed in this work spans three decades, discussing trends over time should be given, for example, has there been a shift in the distribution across countries over the time period.
4. More investigation on how many institutions in each country are working on this, how many different researchers per country – is it just one researcher or group that are contributing to the publication count for a country?
5. In Section 3.4, are the total number of authors unique authors? This should be made clearer. An investigation should be given into collaborations and links between authors. Figure 5 gives the authors who have published the most but are those authors who are tied just working on the same papers?
6. When discussing citation per author in Section 3.4, what do you mean by this? Is this the number of times an author has been cited for their collection of papers that is included in this analysis, that the author has ever been cited in their careers or how often they have been cited within the other papers being analysed? Does that include self-citations?
7. Whilst this is an interesting analysis, it isn’t clear how this research “may help address future research” as stated within the discussion. Providing an overview of the findings of the key papers in the field, e.g. those listed within Table 5, and the general thoughts of the research community on the Nine Hole Peg Test would help with this.
8. Table 5 provides the impact factor of the journals in 2020. However, impact factors change over time and this isn’t reflected in the analysis. The impact factor stated may be quite different from the one which the journal had when the corresponding research paper was published. Giving the impact factor at the time of publication alongside the current impact factor would help with this or alternatively giving a sense of the variation in impact factor for the journal since the corresponding paper was published.
9. In Section 3.5, the authors mention network analysis. The theory of this needs to be briefly discussed and a suitable reference provided. More explanation of the corresponding graphs need to be provided, in particular the significance of the different colouring given in the graphs. If this represents groups, how was this mathematically determined?
10. The data and syntax used to undertake the analysis should be provided for reproducibility and to allow other researchers to see the complete list of publications that have been included within the analysis.
11. Mentions in discussion that there is a growing number of people with the disease. Official statistics need to be provided to support this assertion with a corresponding reference.
Typos/Minor corrections:
1. Line 91 on Page 2: Change to “VOSviewer technique was used to conduct data visualisation”.
2. In Figure 2, change the ordering of the legend such that it reads “1-4” instead of “4-1”, for example.
3. Change “Usa” to “USA”. This occurs several times throughout the manuscript.
4. Table 1 footnote for b has a typo.
5. Typo on line 250 on page 4.
6. Typo on lines 308-309 on page 5.
7. Typo on line 324 on page 6.
Author Response
Dear Mr / Mrs
We welcome your suggestions to improve our work. We have tried to point out all the changes introduced as you will see below. We also attach the document with all the changes introduced according to your recommendations and/or suggestions.
METHOD
A critical point of the analysis is the identification of relevant papers. However, some key terms are not used when undertaking the literature search, e.g. “Nine-Hole Peg Test”, as mentioned within the discussion. However, the exclusion of this term could significantly affect the analysis, in particular as this term is shown as a prominent keyword within the literature, as illustrated within Figure 6. I would strongly recommend extending the literature search to ensure that all relevant papers are identified and included within the analysis.
We decided to select in our search the terms “nine hole peg test” and “nhpt” and “nine hole” and “9hpt” when we have tested that our results did not differ by adding the descriptor "Nine-Hole peg test" in our search. We carried out the search including this descriptor and the results are the same. Within our selection of articles are those that use this key term. An example of this can be found in the article: "Validity of Two Versions of the Nine Hole Peg Test in Stroke Patients Pilot Study( Herren, K and Radlinger, L.) where the term is used in the abstract: " Background: To measure hand dexterity after paresis or injury, in literature different models of the nine-hole-peg test (NHPT) are described".
The data and syntax used to undertake the analysis should be provided for reproducibility and to allow other researchers to see the complete list of publications that have been included within the analysis.
In Material and Methods section, we have included (line 96) a flowchart detailing the syntax used and the process carried out for the search and selection of the included publications.
RESULTS
As the papers being analysed in this work spans three decades, discussing trends over time should be given, for example, has there been a shift in the distribution across countries over the time period.
If we analyse the documents included in this work by decades in terms of their distribution between countries, we observe that in the first decade the vast majority of the studies are carried out in the United States, the country where the instrument was designed and validated. However, in the following two decades, there is an increase in works published by European countries. More specifically, in the second decade we find many works carried out in Germany and Holland, and in the last and most recent decade, Italy also appears as one of the most productive countries. This last data may have to do with the validation of the NHPT as a measure of dexterity in myotonic dystrophy type 1. We have added this comment in the Discussion section(line between 370 and 381).
In Section 3.4, are the total number of authors unique authors? This should be made clearer. An investigation should be given into collaborations and links between authors. Figure 5 gives the authors who have published the most but are those authors who are tied just working on the same papers?
We have changed the order of the sentence so that it can be better understood (line 222 and 223).
As for the comment on figure 5, it shows the most prolific authors of the 615 documents, but table 4 shows the most cited along with their h and g index, where it can be seen that many of them, in turn, are the more prolific.
Table 5 provides the impact factor of the journals in 2020. However, impact factors change over time and this isn’t reflected in the analysis. The impact factor stated may be quite different from the one which the journal had when the corresponding research paper was published. Giving the impact factor at the time of publication alongside the current impact factor would help with this or alternatively giving a sense of the variation in impact factor for the journal since the corresponding paper was published.
We detail the following in methods: "The influence and quality of the journals was also measured using the impact factor obtained from the latest Journal Citation Reports (JCR) (2020) prepared by Clarivate Analytics."(line 110-112). We believe that it is not necessary to put the impact factor of the year of publication of the document since the analysis has been carried out in this year. Perhaps some magazine has changed but we think that it would not be relevant for our study since quantitative data on the use of the NHPT at a specific time are analyzed.
In Section 3.5, the authors mention network analysis. The theory of this needs to be briefly discussed and a suitable reference provided. More explanation of the corresponding graphs need to be provided, in particular the significance of the different colouring given in the graphs. If this represents groups, how was this mathematically determined?
We analysed the most frequently keywords used by authors and the keyword associated with the manuscript by the WOS database following the index of bibliometric studies (Figure 6 and 7, respectively). The keywords most used by the authors and the keyword associated with the manuscript by the WOS database were analyzed following the index of bibliometric studies (Figure 6 and 7, respectively). We have expanded the network of author keywords and plus keywords in the corresponding figures, in addition to making a brief explanation of the characteristics of the figures (between lines 272 and 273 and between lines 276 and 279). VOSviewer technique was used to conduct data visualisation”.
DISCUSSION
When discussing Figure 1, the authors mention a growth in the number of publications from 2008. Some discussion on why this is would aid the reader, e.g. what paper(s) were published that year which created a shift in the research direction?
We have commented in the discussion that this considerable increase in the use of NHPT in research may be due to advances in the study of certain neurodegenerative and/or neurological diseases for their early diagnosis and treatment, as well as the increasing number of people with these diseases, mainly due to the increase in life expectancy of the general population. Most of the articles published in that year tended to deal with issues related to multiple sclerosis and the use of the Multiple Sclerosis Functional Composite (MSFC), a score that combines a measure of lower extremity function (timed walking), extremities superiors (nine hole peg test) and cognitive function (PASAT). Studies claim that it is useful for detecting disability progression in Multiple Sclerosis trials as it is sensitive to changes over time.
Whilst this is an interesting analysis, it isn’t clear how this research “may help address future research” as stated within the discussion. Providing an overview of the findings of the key papers in the field, e.g. those listed within Table 5, and the general thoughts of the research community on the Nine Hole Peg Test would help with this.
We have included a new section about futures researches and Practical or clinical applications. This section will allow to oriented future researches for using a new tool which assess manual dexterity in a widely fields of clinical and research areas. We have included information about results of reliability and validity and applications in neurology disease.
Mentions in discussion that there is a growing number of people with the disease. Official statistics need to be provided to support this assertion with a corresponding reference.
We included the reference in line 308.
Minor corrections:
- Line 91 on Page 2: Change to “VOSviewer technique was used to conduct data visualisation”.
- In Figure 2, change the ordering of the legend such that it reads “1-4” instead of “4-1”, for example.
- Change “Usa” to “USA”. This occurs several times throughout the manuscript.
- Table 1 footnote for b has a typo.
- Typo on line 250 on page 4.
- Typo on lines 308-309 on page 5.
- Typo on line 324 on page 6.
The minor corrections have been modified in the manuscript.

Reviewer 4 Report
The authors attempted to provide a macro narrative of NHPT using a bibliometric approach. Overall, the article has some value with clear charts and graphs, and improves our understanding of NHPT. However, there is obviously space for improvement in this article, especially in terms of depth of research. We suggest the authors to cut down the basic content (knowledge that can be found in the WOS database) as well as add descriptions of research hotspots, frontiers, and trends in the field of NHPT. This could be helpful for scholars to choose their research topics. Sone special comments were provided below.
1-The introduction does not clearly indicate the value and significance of this article. It is suggested to use VOSviewer software.
2-Line 60. One of the research objectives of the article is to understand the "research trajectory" of NHPT, which is difficult to explore through some basic indicators (yearly evolution, country and institution, journal and category distribution, representative authors and article citations). It is recommended that the authors add more content, such as increasing the number of high-frequency keywords, using cluster analysis, and using different views to show the development of the NHPT field. The density view of keywords would allow readers to easily understand which part of the field is focused on research.
3-In terms of search strategy, why did you choose no language restrictions?
4-There is too little detailed description of the study methodology in terms of data analysis and visualization, which makes it very difficult for other researchers to replicate this work. It is recommended to add a description of the VOSviewer software features, as well as the type of analysis, counting methods, and the method of selecting keyword co-occurrence frequencies for this study.
5-Is data de-duplication performed? Because many original studies and conference papers may be the same article. It is better to make a flowchart of data acquisition and pre-processing, which can let readers understand the whole process clearly.
6-Line 321. The authors mention that the search strategy did not include "9-HPT" and "Nine-Hole Peg Test" which may lead to inclusion bias, so why not include these two terms?
7-Line 331. “This study may help address future research related to the use of the NHPT in central nervous system pathologies and those related to the validation of manual dexterity measurement tests.” It is recommended that the value of this study be reorganized and presented in a clearer and more specific way.
Author Response
Dear Mr / Mrs
We welcome your suggestions to improve our work. We have tried to point out all the changes introduced as you will see below. We also attach the document with all the changes introduced according to your recommendations and/or suggestions.
However, there is obviously space for improvement in this article, especially in terms of depth of research. We suggest the authors to cut down the basic content (knowledge that can be found in the WOS database) as well as add descriptions of research hotspots, frontiers, and trends in the field of NHPT. This could be helpful for scholars to choose their research topics. Sone special comments were provided below.
Bibliometric analysis allows to demonstrate a comprehensive overview of the field study of NHPT. This kind of studies focus on: 1) to identify publications and citation trends, 2) to explore the distribution of paper types, 3) to recognize the most relevant countries/regions, affiliations and authors, and 4) to reveal relevant thematic features by analysing publication abstracts and titles with the use of word Keys and topics. The results are relevant because they highlight several research hotspots and emerging topics. In addition, bibliometric studies offer a convenient information for studying collaboration in research, topics, top countries and so on. The results of bibliometric studies provide useful insights into the general status and trends of published scientific literature in relevant areas and the development trends of the areas (Chen, Lun, Yan, Hao, & Weng, 2019). Bibliometric analysis could be complemented by a systematic review for a more extend and meticulous approach including quantitative and qualitative information. However, bibliometrics is the most extensively practiced approach to trace the knowledge anatomy of a research field (Li, Wu, & Wu, 2017) and research topics (Blanco-Mesa, Merigó, & Gil-Lafuente, 2017) while systematic literature reviews are used to synthesize the contents of the literature, limit bias (Tranfield, Denyer, & Smart, 2003). In this work, we have focused on quantitative data and bilbliometric review trying to approach to study area of NHPT. We consider that including a systematic review in the same paper could be excessive, being in itself the object of study for another article. Despite all this, we have included in the article (lines between 393 and 425) a new section with information about clinical and practical applications that could emerge from the results of this study.
The introduction does not clearly indicate the value and significance of this article. It is suggested to use VOSviewer software.
We added more specific information about the studies referenced between lines 39 and 48 and between lines 53 and 59 to improve the introduction section.
We have used the software VOSviewer for the analysis the most frequently used author keywords and plus keywords, specified in the methods section, on the line 114.
One of the research objectives of the article is to understand the "research trajectory" of NHPT, which is difficult to explore through some basic indicators (yearly evolution, country and institution, journal and category distribution, representative authors and article citations). It is recommended that the authors add more content, such as increasing the number of high-frequency keywords, using cluster analysis, and using different views to show the development of the NHPT field. The density view of keywords would allow readers to easily understand which part of the field is focused on research.
In our work we do have the keyword variable considered. Between the lines 272-273 and lines 276-279 we have expanded the keyword information in addition to expanding the number of author and plus keywords in figures 6 and 7, respectively. Indicate that if this variable is contemplated in the objectives, and if some more data can be added about the KEY words in the results, then all the better.
In terms of search strategy, why did you choose no language restrictions?
Most of the articles included in our work are in English and there is a minority in other languages. We consider it interesting not to make a language restriction in our search since the analysis of quantitative indicators allows us to cover publications in different languages that can also be analyzed according to the objective of this article, and that could not be the object of another type of analysis. of content, for example, because it is from other languages that are a minority in scientific publications.
There is too little detailed description of the study methodology in terms of data analysis and visualization, which makes it very difficult for other researchers to replicate this work. It is recommended to add a description of the VOSviewer software features, as well as the type of analysis, counting methods, and the method of selecting keyword co-occurrence frequencies for this study.
In Material and Methods section, we have included (line 96) a flowchart detailing the syntax used and the process carried out for the search and selection of the included publications. The h-index and m-index have also been used for the analysis of author citations(Table 4), as well as the VOSviewer software to analyze the networks of author keywords and keywords plus.
Is data de-duplication performed? Because many original studies and conference papers may be the same article. It is better to make a flowchart of data acquisition and pre-processing, which can let readers understand the whole process clearly.
In Material and Methods section, we have included (line 96) a flowchart detailing the syntax used and the process carried out for the search and selection of the included publications.
The authors mention that the search strategy did not include "9-HPT" and "Nine-Hole Peg Test" which may lead to inclusion bias, so why not include these two terms?
We decided to select in our search the terms “nine hole peg test” and “nhpt” and “nine hole” and “9hpt” when we have tested that our results did not differ by adding the descriptor "Nine-Hole peg test" in our search. We carried out the search including this descriptor and the results are the same. Within our selection of articles are those that use this key term. An example of this can be found in the article: "Validity of Two Versions of the Nine Hole Peg Test in Stroke Patients Pilot Study( Herren, K and Radlinger, L.) where the term is used in the abstract: " Background: To measure hand dexterity after paresis or injury, in literature different models of the nine-hole-peg test (NHPT) are described".
“This study may help address future research related to the use of the NHPT in central nervous system pathologies and those related to the validation of manual dexterity measurement tests.” It is recommended that the value of this study be reorganized and presented in a clearer and more specific way.
Bibliometric analysis allows to demonstrate a comprehensive overview of the field study of NHPT. Our study focused on: 1) to identify publications and citation trends, 2) to explore the distribution of paper types, 3) to recognize the most relevant countries/regions, affiliations and authors, and 4) to reveal relevant thematic features by analysing publication abstracts and titles with the use of word Keys and topics. The results are relevant because they highlight several research hotspots and emerging topics. Then, discussion section shows overall the quantitative data and bilbliometric review trying to approach to study area of NHPT. Other kind of study such as a systematic review will allow us to analyse more qualitative specific content of NHPT. However, we have included in the article (Discussion section) a new section about future researches and clinical or practical applications that could emerge from the results of this quantitative study. This information can be useful for researches and clinical practice in health settings.

Round 2
Reviewer 1 Report
Congratulations to the authors. I think it is an interesting and well-founded work.
Author Response
Dear Mr / Mrs
We appreciate your suggestions to improve our work and your congratulations. The language of the paper has been extensively reviewed by a bilingual native English speaker. If you need a review certificate by the responsible person, do not hesitate to ask us and we will send it to you.
Reviewer 3 Report
Thank you for providing a detailed response to the original comments. Most
comments have been taken into account in the updated manuscript. However, my original comment concerning the graphs in Section 3.5 has not been taken into account, in particular "more explanation of the corresponding graphs need to be provided, in particular the significance of the different colouring given in the graphs. If this represents groups, how was this mathematically determined?" VOSviewer is a software, not a techinque. The mathematics behind the methods used by VOSviewer need to be specified and described briefly in the manuscript with relevant references given.
Author Response
Dear Sir/Madam
We appreciate your suggestions for improving our work. We have tried to point out all changes introduced as you will see below.
In section "2.2 Data analysis and visualization" we have added a reference to the VOSviewer software (line 110) and we have changed the term "technique" to "program".
In section 3.5, between lines 276 and 281, we have provided more information on graphs 7 and 8. These graphs have been produced by the VOSviewer program and the reference provided for it in our work details the mathematical technique used by the software. . Once you enter the data, the program creates the graph with the corresponding colors, depending on the frequency of the terms and their connections.
Reviewer 4 Report
These efforts are more meaningful. I think this article is ready for publication.
Author Response
Dear Mr / Mrs
We appreciate your suggestions to improve our work and your congratulations for our effort. Best regards